# Temperature-dependent responses of the hard corals *Acropora* sp. and *Pocillopora verrucosa* to molecular hydrogen

**Malte Ostendarp**[1]*, **Mareike de Breuyn**[1], **Yusuf C. El-Khaled**[2], **Neus Garcias-Bonet**[2], **Susana Carvalho**[2], **Raquel S. Peixoto**[2], **Christian Wild**[1]

**1** Marine Ecology Department, Faculty of Biology and Chemistry, University of Bremen, Bremen, Germany,
**2** Division of Biological and Environmental Science and Engineering, King Abdullah University of Science and Technology, Thuwal, Saudi Arabia

* maos@uni-bremen.de

## Abstract

Coral reefs are increasingly threatened by mass bleaching events due to global ocean warming. Novel management strategies are urgently needed to support coral survival until global efforts can mitigate ocean warming. Given the strong antioxidant, anti-inflammatory and anti-apoptotic properties of molecular hydrogen, our study explores its potential to alleviate the negative effects of heat stress on corals. We investigated the ecophysiological responses of two common hard corals (*Acropora* sp. and *Pocillopora verruco*sa) from the Central Red Sea under ambient (26 °C) and elevated seawater temperatures (32 °C), with and without hydrogen addition (~ 150 µM H$_2$) over 48 h. Our results showed that at 32 °C without hydrogen addition, *P. verrucosa* exhibited high temperature tolerance, whereas *Acropora* sp. showed significant reductions in photosynthetic efficiency and maximum electron transport rate compared to the ambient condition (26 °C). The addition of hydrogen at 32 °C increased the maximum electron transport rate of *Acropora* sp. by 28%, maintaining it at levels compared to those at 26 °C. In contrast, the addition of hydrogen at 26 °C caused a significant decrease in the photophysiology of both *Acropora* sp. and *P. verrucosa*. This suggests that the short-term response of the coral holobiont to molecular hydrogen is temperature-dependent, potentially benefiting the coral holobiont under heat stress, while impairing the photophysiology under ambient temperatures. Our findings therefore provide the foundation for future long-term studies uncovering the mechanisms behind molecular hydrogen, potentially informing the development of new management strategies to enhance coral resilience to ocean warming.

## Introduction

Coral reefs are among the most biodiverse and productive ecosystems worldwide [1]. They provide humankind with a diverse range of essential ecosystem services [1], on which millions of livelihoods depend [2]. However, coral reefs are currently in serious decline [3,4], as a result of many different local and global anthropogenic and natural threats, including local pollution, overfishing, and global warming [5–9].

**Data availability statement:** All relevant data are within the paper and its Supporting information files.

**Funding:** MO and CW were supported by baseline funds from the University of Bremen and DFG grant Wi 2677/24-1. RSP was supported by KAUST grant BAS/1/1095-01-01. The funders had no role in study design, data collection and analysis, decision to publish, or preparation of the manuscript.

**Competing interests:** The authors have declared that no competing interests exist.

Although many of these threats are currently identified and partially addressed, climate change continues at a rapid pace with minimal progress made towards mitigation [10,11]. In association with the onset of El Niño in 2023 [12], new ocean temperature records are emerging [13,14], greatly increasing the probability and frequency of mass coral bleaching and mortality events worldwide [10,15–17]. Direct actions and strategies are urgently needed to further prevent the rapid degradation of these highly valuable ecosystems and buy time for coral reefs until climate change becomes manageable by global efforts [18].

Climate change-associated shifts in physical and biogeochemical conditions are widely recognized to induce coral bleaching, which is the loss of the coral's endosymbiotic algae (Symbiodiniaceae) that often represent the main energy source of the coral host [19,20]. Novel intervention approaches could therefore aim to target physiological cascades associated with coral bleaching, particularly those initiated by heat stress, offering a promising approach for improving coral resilience. Several prominent cascades have been proposed, highlighting the roles of both nitrogen availability and oxidative stress [20]. One cascade involves the disruption of a nitrogen-limited state, essential for maintaining a stable symbiosis between the coral host and its algae, by an increased nitrogen availability [19,21,22]. Another cascade involves the increased generation of reactive oxygen species (ROS) and reactive nitrogen species (RNS), primarily triggered by temperature and light stress, that overwhelms the coral's antioxidative capacity and potentially induces a stress response ultimately resulting in coral bleaching as proposed by the "Oxidative Theory" [23,24].

In this context, molecular hydrogen might offer an effective treatment to mitigate coral bleaching. Due to hydrogen's strong antioxidant, anti-inflammatory and anti-apoptotic characteristics, it has been used as a preventive and therapeutic agent in human medicine [25–30]. As a strong antioxidant, molecular hydrogen selectively reduces hydroxyl radicals and peroxynitrite, which are among the most harmful ROS/RNS [25]. In contrast, other ROS/RNS with a potentially beneficial physiological role are not affected by molecular hydrogen [25,29,30]. Since molecular hydrogen is also able to rapidly diffuse across membranes, it can successfully penetrate cell organelles [25] and scavenge ROS/RNS inside the cytoplasm and mitochondria. In addition, molecular hydrogen is proposed to inhibit the generation of ROS/RNS by preventing leakage of electrons from the electron transport chain [30], regulate gene expression [29], increase overall antioxidant capacity [31,32] and upregulate the heat shock response [33].

Based on these previously observed effects of molecular hydrogen in mammals including humans and rats [25–30], here we provide first insights into the potential effects of molecular hydrogen on the coral holobiont. Our objective was to assess if molecular hydrogen affects the ecophysiology of corals under heat stress. Considering its antioxidant, anti-inflammatory and anti-apoptotic properties [25–30], we hypothesize that molecular hydrogen minimizes negative effects of heat stress.

To address the research question and associated hypothesis, we experimentally examined the short-term effect (48 h) of hydrogen-enriched seawater under ambient temperature and heat stress on the two common and widely distributed hard corals *Acropora* sp. and *P. verrucosa* from the Central Red Sea. For this purpose, we analyzed several physiological parameters related to bleaching phenotype, photosynthetic capacity and holobiont phenotype. For the bleaching phenotype, Symbiodiniaceae cell density, chlorophyll a and c2 concentrations and primary production ($P_{net}$ and $P_{gross}$) were analyzed. These parameters can directly indicate a loss of endosymbionts and pigmentation, which are associated with reduced primary production and serve as key indicators of coral bleaching [34].

Building on the assessment of the bleaching phenotype, we further evaluated the photosynthetic capacity by measuring the photosynthetic efficiency, maximum electron transport

rate ($ETR_{max}$), and minimum saturating irradiance ($E_k$) with $ETR_{max}$ indicating the maximum capacity of the electron transport chain and $E_k$ reflecting the light intensity at which photosynthesis becomes saturated [35]. These parameters therefore provide insight into the functional state of the photosynthetic apparatus [35], with temperature-induced shifts indicating the loss of healthy endosymbionts or revealing disruptions within the photosynthetic machinery, even before the loss of endosymbionts occurs [36,37].

At the holobiont phenotype, we assessed respiration rates (R), which serve as a proxy for the metabolic demand typically increasing under stress [34]. By connecting these parameters, we generated a comprehensive overview of coral holobiont health, identifying the physiological responses of *Acropora* sp. and *P. verrucosa* to short-term heat stress and evaluating the potential role of hydrogen in minimizing temperature-induced negative effects.

## Materials and methods

### Coral collection and experimental preparation

To assess the short-term effect of hydrogen addition on the coral holobiont, two common Red Sea coral species, *Acropora* sp. and *P. verrucosa*, were selected. *Acropora* fragments were collected based on morphological characteristics (*Acropora* cf. *hemprichii*). However, due to the complex taxonomy within the genus *Acropora*, we decided to conservatively refer to *Acropora* sp. Coral fragments were sampled by SCUBA diving near the "Coral Probiotics Village" at Al Fahal Reef (22°18'19.1"N, 38°57'55.0"E), located about 20 km off the King Abdullah University of Science and Technology, Thuwal, in the Saudi Arabian coast of the Red Sea. Permits were obtained from the Institutional Biosafety and Bioethics Committee of the King Abdullah University of Science and Technology (IBEC protocol number 22IBEC003_v4).

A total of six colonies per species were tagged at a depth of approximately 10 m to allow the identification and resampling of the same colonies during the experiment duration. On each sampling day, two coral fragments per colony measuring about 3 - 5 cm in length were collected using pliers along with 90 L of seawater in water containers. Due to laboratory space limitations, it was not possible to run all incubations simultaneously. Therefore, short-term incubations (48 h) were carried out on four different days, with each day randomly assigned to a specific temperature and treatment group (CT26, CT32, $H_2$26, $H_2$32). These incubations were conducted within the shortest possible timeframe lasting from 07th to 23rd of March 2023 to minimize the impact of potential seasonal changes. For each incubation, the respective coral fragments were collected on the same day. Our decision for this approach was driven by the need for controlled environmental conditions, the feasibility of treatment application, and the aim to capture a large biological variability of corals. Consequently, our setup was restricted to one tank per treatment and species, as it was intended to apply heat stress and treatments over a 48-h period rather than for extended coral maintenance. Reef environmental parameters, including seawater temperature and salinity, were recorded during this period (S1 File) using a multiparameter conductivity, temperature and depth (CTD) sensor (Ocean Seven 310, Idronaut, Italy).

### Experimental design

After collection, the coral fragments were transported to the laboratory in transparent sampling bags within a plastic box containing ambient seawater and a portable aquarium pump to ensure aeration. Immediately after arrival, all 24 fragments were attached to custom-made stands marked with colored stripes for the respective colony and placed inside two plastic aquaria, one for each species, filled with 30 L of seawater collected from the reef. The aquaria were equipped with an aquarium pump and a heater connected to a thermostat while natural

lighting was simulated with a LED lamp replicating a 12 h light/dark-cycle (PAR ~ 200 μmol m$^{-2}$ s$^{-1}$, salinity 39.0 ± 1.0 ‰, temperature ambient 26.0 ± 0.5 °C/ increased 32 ± 0.5 °C). On each sampling day, all fragments were incubated for 48 h in one treatment (CT = control, H$_2$ = hydrogen-enriched) and under one temperature condition (ambient = 26 °C, heat stress = 32 °C) resulting in four different groups: CT26, CT32, H$_2$26, H$_2$32.

To increase the hydrogen concentration, a hydrogen water generator was used (Hydrogen-rich Water Cup, ABS-FQ-02, Aukewel) that produced 240 mL of hydrogen-enriched distilled water with a concentration of ~ 3 ppm each cycle according to test reports provided by the manufacturer. In all groups, 10% of the total seawater was exchanged at the start and after 24 h with artificial seawater created with MilliQ-water and marine sea salt (Red Sea salt, Red Sea) at the respective temperature. In the hydrogen treatments, the artificial seawater was previously hydrogen enriched resulting in a theoretical hydrogen concentration of ~ 0.3 ppm (~ 150 μM). However, as molecular hydrogen is barely soluble [29] and is constantly released, we can only assume a hydrogen spike at the start and after 24 h of the experiment. After the first water exchange, the temperature was ramped in the heat stress groups from 26 °C to 32 °C within 2 h. Temperature and salinity were checked consistently throughout the experiment using a handheld sensor (YSI professional plus handheld multiparameter meter) to potentially adjust temperature and salinity. Due to a malfunctioning heater in the *P. verrucosa* aquarium during the CT32 group, the water temperature decreased in the first 12 h after the ramping to 28 °C and was then ramped up back to 32 °C within 2 h.

Following the 48 h incubation period, several response parameters related to coral physiology and the overall coral holobiont health were assessed as proxies of the stress state. Survival and bleaching of every fragment were visually analyzed. Then, six fragments per species were used to measure photosynthesis and respiration rates, photosynthetic efficiency (*Fv/Fm*) and rapid light curves (RLC). The remaining six fragments per species were frozen for further analyses of Symbiodiniaceae cell density and chlorophyll a and c2 content. To normalize response parameters to the size of the coral fragments, the surface area was measured using 3D models in Autodesk ReCap Photo (version 23.1.0) according to Lavy et al. [38] and Tilstra et al. [39]. For this purpose, at least 20 photographs were taken from all sides of the fragment. Photographs were taken after the incubation to avoid stressing the corals beforehand. From these photographs 3D models were created and sliced to only include the fragment surface. The surface area in cm$^2$ was then calculated by the program using a reference scale placed on each picture.

## Data collection

**Survival and bleaching.** Survival and bleaching of all fragments were visually assessed directly after the 48 h incubations. Fragments were classified as dead if 100% tissue loss was visible.

**Photosynthesis and respiration.** Six fragments per species were placed on custom-made stands into 1 L jars filled with the respective treatment water and sealed airtight without any air bubbles inside to measure oxygen fluxes according to Tilstra et al. [39] and Mezger et al. [40]. In addition, three 1 L jars per aquaria were filled with treatment water excluding any coral fragments to account for background photosynthesis and respiration rates. Briefly, all jars were placed into a water bath equipped with a heater connected to a thermostat to maintain the temperature at the respective temperature condition. Magnetic stirrers (~ 200 rpm) placed onto stirring plates (magnetic stirrer HI 200M, Hanna instruments) ensured a constant water flow and homogenous conditions inside the jars. Initially, all fragments and seawater blanks were light-incubated for 90 min (PAR ~ 200 μmol m$^{-2}$ s$^{-1}$) and afterwards

dark-incubated for 60 min. The variation in light and dark incubation period ensured that the oxygen concentration remained within a reliable range. Oxygen concentrations were measured at the start and end of the light and dark incubation using an oxygen probe (Multi 3500i handheld multimeter, WTW and YSI professional plus handheld multiparameter meter). Two calibrated oxygen probes were used in between treatments due to technical difficulties (Multi 3500i handheld multimeter, WTW was used for the $H_2$26 group and YSI professional plus handheld multiparameter meter for the CT26, CT32 and $H_2$32 groups).

To calculate net photosynthesis ($P_{net}$) and respiration (R) rates, start and end oxygen concentrations of the light and dark incubations were subtracted, respectively. $P_{net}$ and R were then normalized by the surface area of each fragment, total volume of the incubation jar, total incubation time and the background photosynthesis and respiration rates.

Using $P_{net}$ and R, the gross photosynthesis ($P_{gross}$) was then calculated according to following formula:

$$P_{gross} = P_{net} + |R|$$

**Photosynthetic efficiency and rapid light curves.** Photosynthetic efficiency was measured directly after the dark incubation using an Imaging-PAM (model IMAG-K7, Walz GmbH) according to Ralph and Gademann [35] and Ralph et al. [41], as the corals were already dark-adapted for > 30 min. For each fragment, the dark level fluorescence yield (*F0*) and the maximum fluorescence yield (*Fm*) were measured using the following PAM settings: gain = 2, damp = 2, saturation intensity = 7, saturation width = 4 and measuring light intensity = 7. The optimal PSII quantum yield (*Fv/Fm*) was calculated for five points randomly placed on the coral fragment. In addition, a rapid light curve (RLC) was generated afterwards for the same five points through an exposure of twelve light-series with increasing irradiance from 0 to 702 µmol photons $m^{-2} s^{-1}$. Between the twelve light-series, the fragments were illuminated with the respective actinic light for 20 s, each ended by a saturation pulse. Through all of the five RLCs per fragment one model was fitted according to Platt et al. [42] using the following equation:

$$y = ps \times \left(1 - e^{\frac{-x \times \alpha}{ps}}\right) \times e^{\frac{-x \times \beta}{ps}}$$

For the model, values of 0 µmol electrons $m^{-2} s^{-1}$ which were recorded at higher irradiances than 0 µmol photons $m^{-2} s^{-1}$ were excluded in advance. This exclusion was made due to the model's sensitivity as the variables of interest were chosen prior to this range. To determine the variables ps, α and β, a pseudo-random search algorithm was run within the package *phytotools v.1.0* [43] in the program R *v.4.3.0* [44]. The maximum electron transport rate ($ETR_{max}$) and minimum saturating irradiance ($E_k$) were calculated using the determined variables as follows:

$$ETR_{max} = ps \times \frac{\alpha}{\alpha + \beta} \times \left(\frac{\beta}{\alpha + \beta}\right)^{\frac{\beta}{\alpha}}$$

$$E_k = \frac{ETR_{max}}{\alpha}$$

**Symbiodiniaceae cell densities and chlorophyll concentrations.** Coral tissue was removed using an airbrush filled with 10 mL distilled water. The tissue slurry was collected in a 15 mL falcon tube to measure the total volume. In between fragments, the airbrush was

cleaned with 70% ethanol. After airbrushing, the slurries were frozen at -20 °C until further analysis.

To assess the Symbiodiniaceae cell density, a total of 100 µL tissue slurry was thawed and vortexed (analog vortex mixer, VWR). Each sample was then centrifuged for 5 min at 8000 rpm (centrifuge 5424R, Eppendorf). Subsequently, the supernatant was discarded and 1 mL of MilliQ-water was added to the pellet. After dissolving the pellet and homogenizing the mixture using a vortex (analog vortex mixer, VWR), the sample was pipetted on a cell strainer with a 30 µm mesh size to remove larger cells. Following filtration, 200 µL per sample were placed in duplicates in a microplate well. Symbiodiniaceae cell density was then measured using a flow cytometer (Accuri C6 Flow Cytometer, BD). Between samples, a washing and agitating cycle was run and in between three samples, the plate was vortexed to ensure a homogenous mixture. Each sample was run for 2 min and if the events/s recorded by the flow cytometer exceeded 1000, the sample was diluted with MilliQ-water and rerun. The initial count was normalized to the volume sampled by the flow cytometer, surface area, dilution and slurry volume.

A total of 1 mL tissue slurry was thawed and used to measure chlorophyll a and c2 in duplicates. Each sample was centrifuged (centrifuge 5424R, Eppendorf) for 5 min with 5000 rpm at 4 °C and afterwards the supernatant was discarded. To extract chlorophyll a and c2, 2 mL of 90% acetone was added to each sample. The samples were then sonicated for 10 min and dark-incubated for 24 h at 4 °C. Following 24 h, 200 µL of each sample were transferred in duplicates in a micro-plate well. Additionally, three 200 µL 90% acetone blanks were placed on each microplate well to normalize values afterwards. The chlorophyll a and c2 concentrations were assessed by measuring the absorbance at 630 nm ($E_{630}$) and 663 nm ($E_{663}$) using a spectrophotometer (Multiskan SkyHigh microplate spectrophotometer v.1.0.70.0, Thermo Fisher Scientific). Final values were calculated according to Jeffrey and Humphrey [45] and normalized to the surface area, the slurry volume and the averaged absorbance of the blanks. As Jeffrey and Humphrey [45] assumed a path length of 1 cm path which is inaccurate for a microplate well, all values were additionally divided by 0.555. Final chlorophyll a and c2 concentrations were also normalized by Symbiodiniaceae cell density.

**Statistical analyses.** The statistical analyses were conducted in the PRIMER-E software *v.6.1.18* [46] with the PERMANOVA + add on *v.1.0.8* [47] using a permutational multivariate analysis of variance (PERMANOVA). The type III (partial) PERMANOVAs with unrestricted permutations of raw data (999 permutations) including a Monte Carlo test were performed on a resemblance matrix created with the Euclidean distance of the previously normalized data. Two-factorial (factors: temperature [T; two levels: 26 °C, 32 °C], treatment [Tr; two levels: Control, Hydrogen]) PERMANOVAs were carried out for each species on the variables within the oxygen fluxes ($P_{net}$, R, $P_{gross}$), photosynthetic efficiency (*Fv/Fm*), rapid light curves (α, ps/ $ETR_{max}$, $E_k$), chlorophyll ($Chl_a$, $Chl_{c2}$) and Symbiodiniaceae cell density (S1 File). The results were considered significant below a p-value of 0.05. If the interaction effect (T x Tr) was significant, PERMANOVA pairwise tests including a Monte Carlo test were conducted within the temperature and treatment level.

The RLC models and all graphs were created in R *v.4.3.0* [44] using the packages *phytotools v.1.0* [43], *tidyverse v.2.0.0* [48], *ggpubr v.0.6.0* [49] and *rstatix v.0.7.2* [50]. All plots and values represent the mean value and the respective standard error (MEAN ± SE). The raw data is provided in the supporting information (S1 Data).

# Results

## Survival and bleaching

Neither *Acropora* sp. nor *P. verrucosa* exhibited visual signs of mortality or bleaching across all treatments (CT26, CT32, $H_2$26, $H_2$32).

## Photosynthesis and respiration

Gross photosynthesis and respiration rates of *Acropora* sp. were significantly affected by temperature ([Fig 1A](), PERMANOVA, p (respiration) = 0.004, p (gross photosynthesis) = 0.049). Under heat stress (CT32 and $H_2$32), gross photosynthesis and respiration rates were significantly increased compared to ambient seawater temperatures (CT26 and $H_2$26).

In contrast, net and gross photosynthesis as well as respiration rates of *P. verrucosa* were significantly affected by an interaction between treatment and temperature ([Fig 1B](), PERMANOVA, p (respiration) = 0.001, p (net photosynthesis) = 0.019, p (gross photosynthesis) = 0.003). The addition of hydrogen at ambient seawater temperatures ($H_2$26) caused a significant decline in net photosynthesis, gross photosynthesis, and respiration rates of *P. verrucosa* compared to CT26 ([Fig 1B](), pairwise PERMANOVA, p (respiration) = 0.013, p (net photosynthesis) = 0.001, p (gross photosynthesis) = 0.003). However, the addition of hydrogen under heat stress ($H_2$32) resulted in significantly higher respiration, net photosynthesis, and gross photosynthesis rates of *P. verrucosa* compared to $H_2$26 ([Fig 1B](), pairwise PERMANOVA, p (respiration) = 0.001, p (net photosynthesis) = 0.011, p (gross photosynthesis) = 0.001).

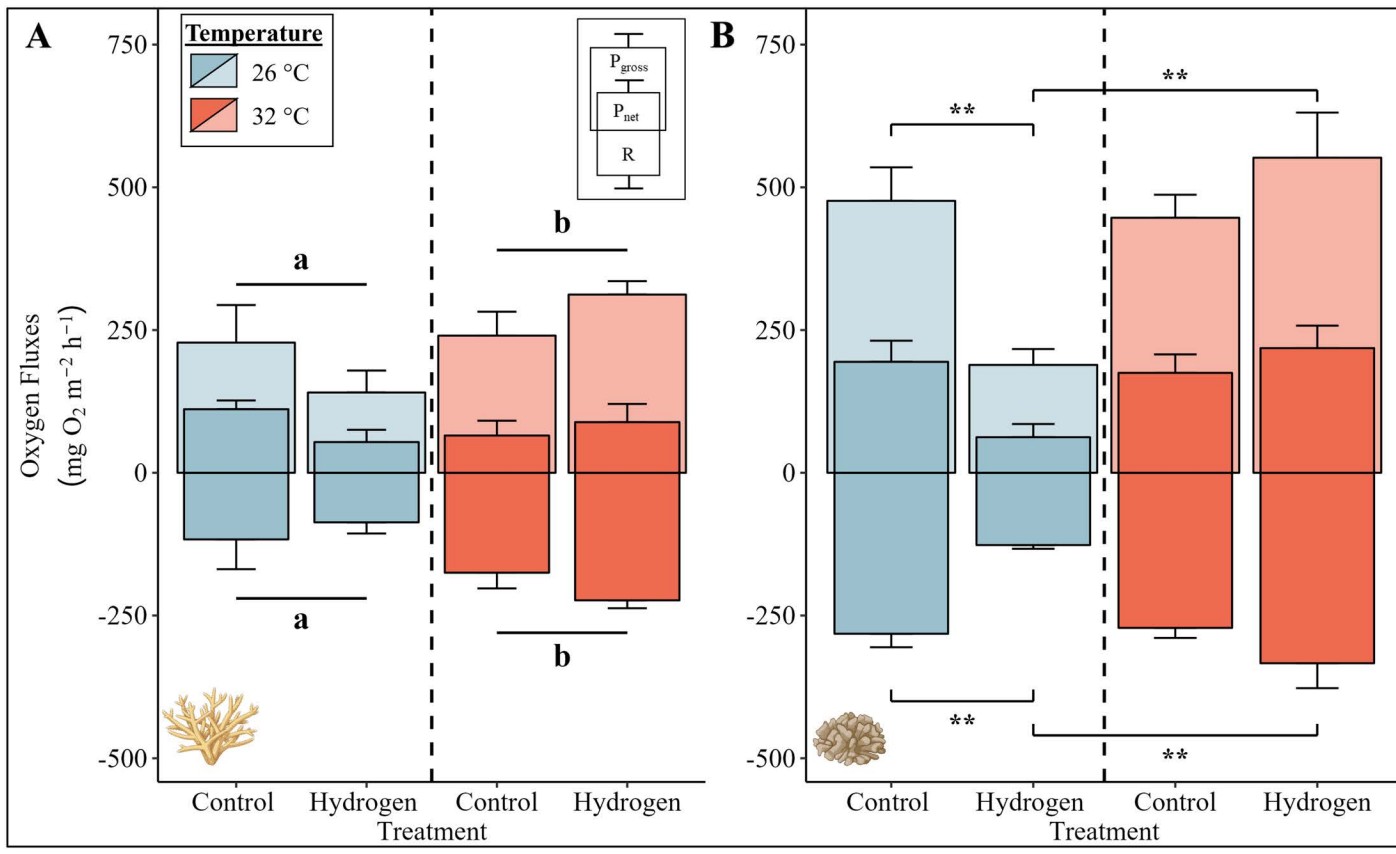

**Fig 1. Photosynthesis and respiration rates of (A)** *Acropora* **sp. and (B)** *P. verrucosa* **for the four groups.** Mean net and gross photosynthesis ($P_{net}$ and $P_{gross}$) and respiration (R) rates are visualized by bars with error bars indicating the respective standard error (SE) of six replicates per group (except for four in the CT26 group *Acropora* sp. and five in the CT32 group *P. verrucosa* due to a measurement error). The color of the bars indicates the temperature condition (blue = 26 °C, red = 32 °C). For *Acropora* sp. (A), a significant temperature main effect according to the PERMANOVA with Monte-Carlo test is presented with letters ([a/b] p < 0.05; above bar: $P_{gross}$, below bar: R) and black lines grouping the temperature conditions. For *P. verrucosa* (B), pairwise PERMANOVAs with Monte-Carlo tests were conducted as the interaction effect between temperature and treatment was significant. Significant differences in between treatments are illustrated with asterisks (* p < 0.05, ** p < 0.01; above bar: $P_{gross}$, below bar: R). Exact p-values are presented in supplementary material ([S1 File]()). Coral illustrations were created with BioRender.com.

These photosynthesis and respiration rates at $H_2 32$ were resembling those observed in both control groups at ambient (CT26) and heat stress (CT32) conditions.

## Photosynthetic efficiency and rapid light curves

Similar to the photosynthesis and respiration rates, the photosynthetic efficiency of *Acropora* sp. was significantly affected by temperature (Fig 2A, PERMANOVA, p = 0.003). Heat stress (CT32 and $H_2 32$), however, caused a significant reduction in the photosynthetic efficiency in comparison to the treatments without heat stress (CT26 and $H_2 26$).

In comparison, the photosynthetic efficiency of *P. verrucosa* was not affected by temperature or treatment.

The $ETR_{max}$ and $E_k$ of *Acropora* sp., derived from the RLCs (Fig 3A), were both significantly affected by an interaction between temperature and treatment (Fig 3B and 3C, PERMANOVA, p ($ETR_{max}$) = 0.002, p ($E_k$) = 0.019). $ETR_{max}$ of *Acropora* sp. significantly decreased in the heat stress treatment CT32 (Fig 3B, pairwise PERMANOVA, p = 0.003) and in the hydrogen treatment $H_2 26$ (Fig 3B, pairwise PERMANOVA, p = 0.001) compared to the control treatment CT26. However, $ETR_{max}$ remained significantly higher for *Acropora* sp. in the combined treatment of heat stress and hydrogen addition ($H_2 32$) than in the CT32 (Fig 3B, pairwise PERMANOVA, p = 0.044) and $H_2 26$ treatments (Fig 3B, pairwise PERMANOVA, p = 0.036) and comparable to the $ETR_{max}$ observed in the CT26 treatment. Similar to the decrease in the $ETR_{max}$, $E_k$ values of *Acropora* sp. also declined significantly in the presence of hydrogen-enriched seawater ($H_2 26$) compared to the control group CT26 (Fig 3C, pairwise PERMANOVA, p = 0.008).

While the $ETR_{max}$ of *P. verrucosa*, derived from the RLCs (Fig 4A), remained unaffected by heat stress or hydrogen addition (Fig 4B), $E_k$ was significantly affected by temperature (Fig 4C, PERMANOVA, p = 0.018) causing a significant decrease in both heat stress treatments (CT32 and $H_2 32$) compared to the treatments without heat stress (CT26 and $H_2 26$).

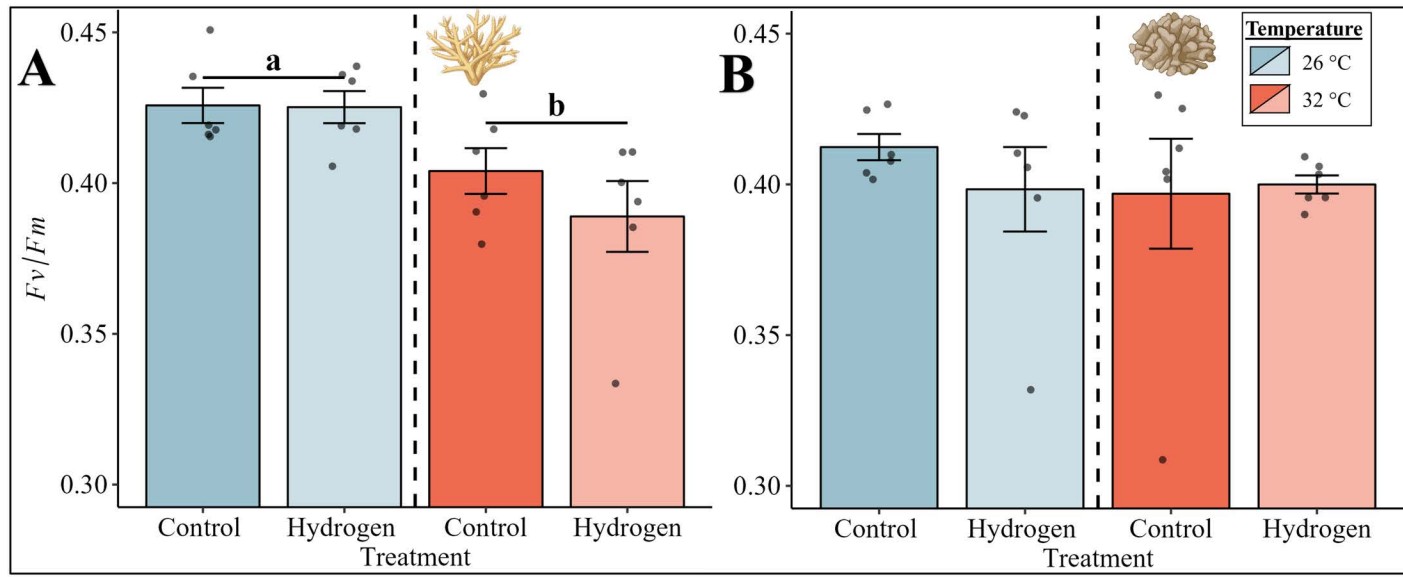

**Fig 2. Photosynthetic efficiency (*Fv/Fm*) of (A) *Acropora* sp. and (B) *P. verrucosa* for the four groups.** Mean photosynthetic efficiency is visualized by bars with error bars indicating the respective standard error (SE) of six replicates per group. The color of the bars indicates the temperature condition (blue = 26 °C, red = 32 °C). A significant temperature main effect according to the PERMANOVA with Monte-Carlo test (p < 0.05) is presented with letters and black lines grouping the temperature conditions. Exact p-values are presented in supplementary material (S1 File). Coral illustrations were created with BioRender.com.

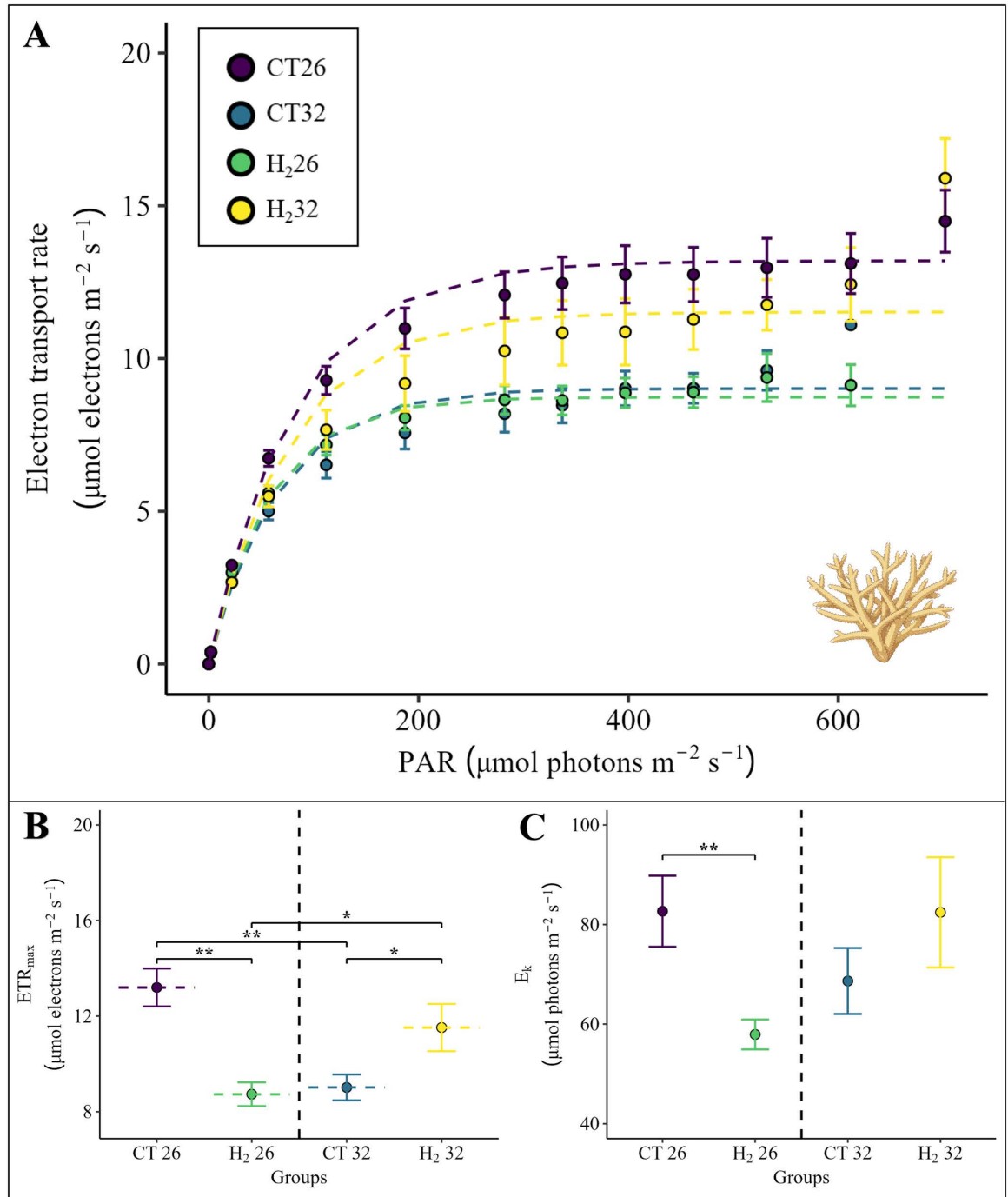

**Fig 3. (A) Rapid light curves, (B) maximum electron transport rates (ETR$_{max}$) and (C) minimum saturating irradiances (E$_k$) derived from the model fitted according to Platt et al. [42] for all four groups belonging to** *Acropora* **sp.** The rapid light curves and the respective ETR$_{max}$ and E$_k$ values are calculated as the mean of six replicates per group (S1 File). The four groups are illustrated with different colors. Points indicate the mean value and error bars the respective standard error (SE). Pairwise PERMANOVAs with Monte-Carlo tests were conducted if the interaction effect between temperature and treatment was significant and are illustrated with asterisks indicating a significant difference ( * p < 0.05, ** p < 0.01). Exact p-values are presented in supplementary material (S1 File). Coral illustration was created with BioRender.com.

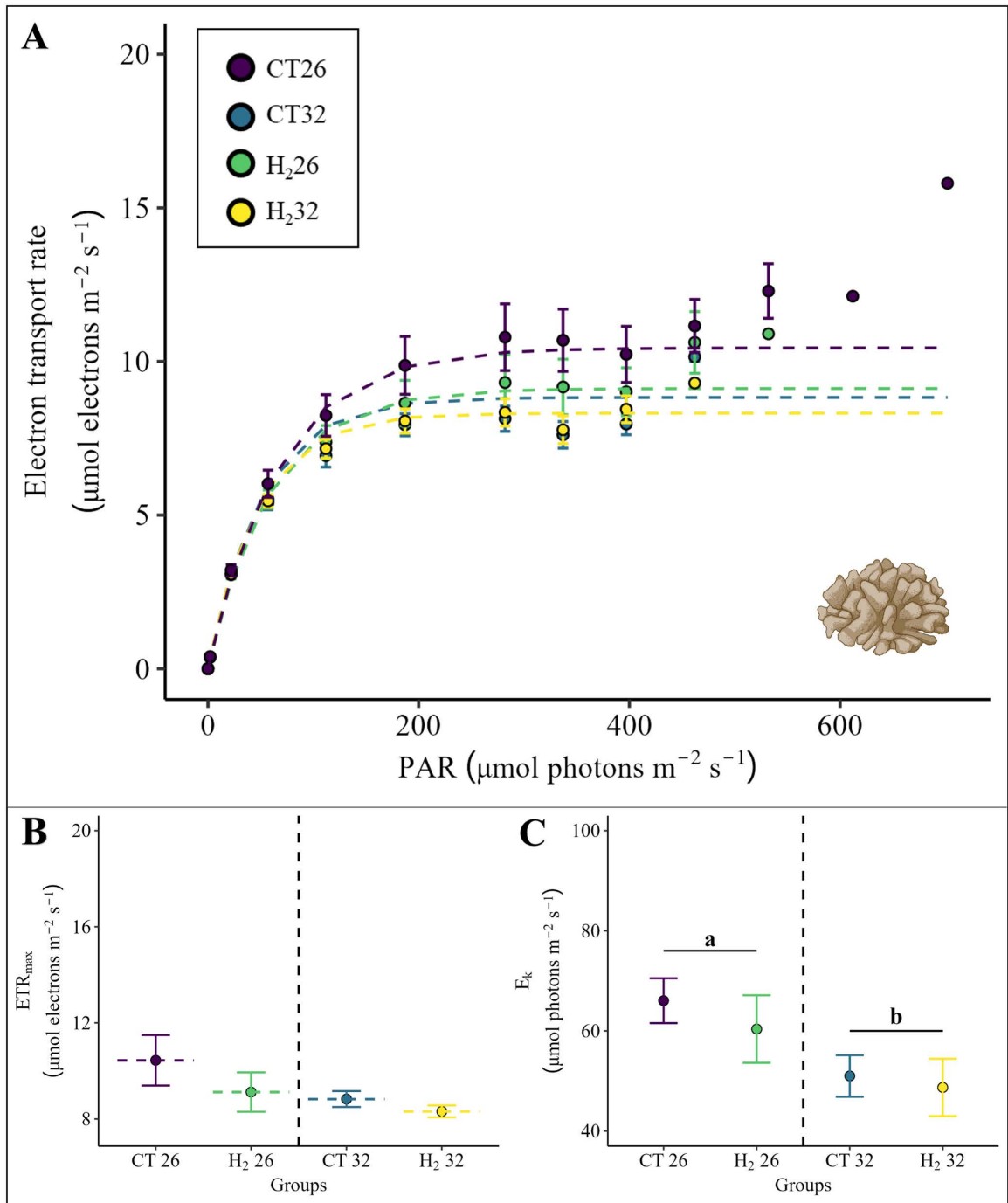

**Fig 4. (A) Rapid light curves, (B) maximum electron transport rates (ETR$_{max}$) and (C) minimum saturating irradiances (E$_k$) derived from the model fitted according to Platt et al. [42] for all four groups belonging to** *P. verrucosa*. The rapid light curves and the respective ETR$_{max}$ and E$_k$ values are calculated as the mean of six replicates per group (S1 File). The four groups are illustrated with different colors. Points indicate the mean value and error bars the respective standard error (SE). A significant temperature main effect according to the PERMANOVA with Monte-Carlo test ($p < 0.05$) is presented with letters and black lines grouping the temperature conditions. Exact p-values are presented in supplementary material (S1 File). Coral illustration was created with BioRender.com.

## Symbiodiniaceae cell densities and chlorophyll concentrations

The Symbiodiniaceae cell density of *Acropora* sp. was significantly affected by temperature (Fig 5A, PERMANOVA, p = 0.047). Resembling the significant decline in photosynthetic efficiency, Symbiodiniaceae cell density of *Acropora* sp. decreased in both heat stress treatments (CT32 and $H_2$32) in comparison to the treatments at ambient temperature (CT26 and $H_2$26). However, chlorophyll a and c2 concentrations remained unaffected by temperature or treatment (Fig 5C and 5E).

While the Symbiodiniaceae cell density and chlorophyll a concentration of *P. verrucosa* was not significantly affected by temperature or treatment (Fig 5B and 5F), the areal chlorophyll $c_2$ concentration showed a significant effect of the hydrogen treatment (Fig 5D, PERMANOVA, p = 0.039). The addition of hydrogen ($H_2$26 and $H_2$32) resulted in a significant decrease of the areal chlorophyll $c_2$ concentration in comparison to both treatments without hydrogen addition (CT26 and CT32).

## Discussion

Coral reefs are currently experiencing the fourth global coral bleaching event [17,51], reaffirming the projections for an increase in both the scale and frequency of such events [10,15–17]. It is therefore crucial to deepen our understanding of the mechanisms driving coral bleaching [20] and to investigate new approaches to strengthen coral reefs, thereby potentially counteracting these events in the future [52,53]. In this context, molecular hydrogen might offer a promising approach since several studies have already demonstrated an extensive preventive and therapeutic effect of hydrogen in mammals including humans and rats [25–30]. Our study is the first to investigate the short-term effects of hydrogen on the ecophysiology of the coral holobiont under ambient temperatures and heat stress, providing insight into potential ecological implications.

### Molecular hydrogen can minimize negative effects of heat stress on the hard coral *Acropora* sp.

When under heat stress alone, *Acropora* sp. experienced a significant reduction in photosynthetic efficiency, $ETR_{max}$ and Symbiodiniaceae cell density alongside an increase in metabolic demand - indicative of a pronounced heat stress response [54,55]. However, simultaneous exposure to hydrogen and heat stress prevented the decline in the $ETR_{max}$ observed with heat stress alone in *Acropora* sp. As shown in previous studies, a decrease in $ETR_{max}$ can serve as an indicator of photosystem damage and a deteriorated health state of the Symbiodiniaceae cells [36,37,56]. This damage likely originates from the ROS/RNS accumulation under heat stress as proposed by the "Oxidative Theory" [20,23,24,57,58]. Our results therefore suggest that molecular hydrogen might have protected the coral holobiont from oxidative damage, thereby preventing a decrease in $ETR_{max}$. These findings align with the previously reported antioxidative capacity of molecular hydrogen [25–30]. In addition, molecular hydrogen may protect the photosynthetic apparatus indirectly by increasing antioxidant enzyme activity [59,60] and upregulating the heat shock response as observed in plants and rodents, respectively [33].

Consistent with our observations, a previous study in the Central Red Sea reported that *Acropora* sp. was more affected by simulated short-term heat stress than *P. verrucosa* [61]. This different response of *P. verrucosa* could indicate that the simulated heat stress was insufficient to induce negative effects within 48 h and suggests a higher temperature resistance in *P. verrucosa*, even though other studies have described *P. verrucosa* as more sensitive to temperature stress than *Acropora hemprichii* in the Red Sea [62]. The difference in the thermotolerance of the two investigated species may also be linked to the endosymbionts associated, as in

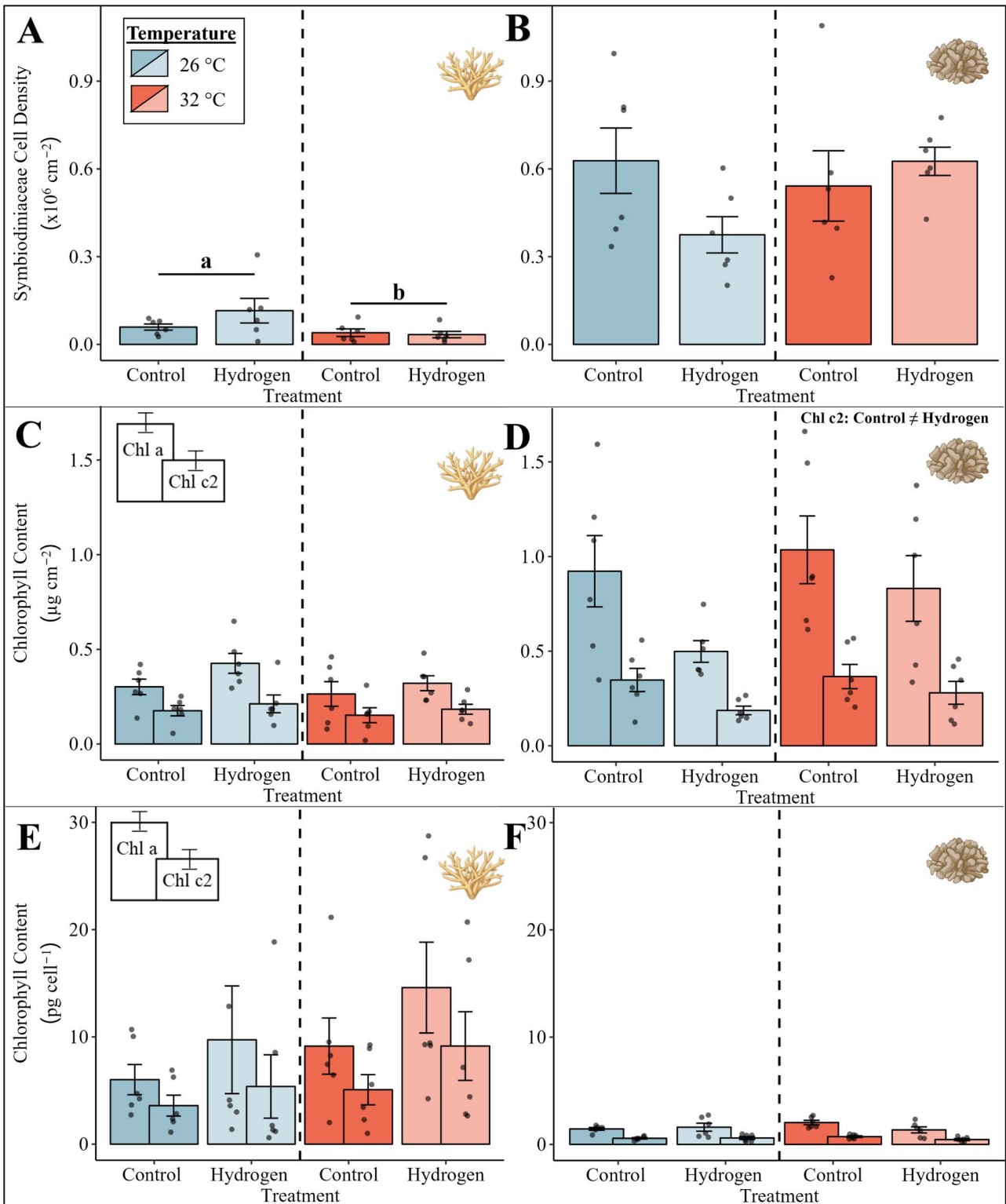

**Fig 5. Symbiodiniaceae cell density of (A)** *Acropora* **sp. and (B)** *P. verrucosa*, **areal chlorophyll content of (C)** *Acropora* **sp. and (D)** *P. verrucosa*, **and chlorophyll content per cell of (E)** *Acropora* **sp. and (F)** *P. verrucosa* **for the four groups.** Bars with error bars indicate the mean and the respective standard error (SE) of six replicates per group. For the chlorophyll content, chlorophyll a is presented on the left side of each bar with chlorophyll c2 at the right side. The color of the bars indicates the temperature condition (blue = 26 °C, red = 32 °C). A significant temperature main effect according to the PERMANOVA with Monte-Carlo test ($p < 0.05$) is presented with letters and black lines grouping the temperature conditions. A

significant treatment main effect according to the PERMANOVA with Monte-Carlo test (p < 0.05) is presented with the unequal sign in the top right corner. Exact p-values are presented in supplementary material (S1 File). Coral illustrations were created with BioRender.com.

the Red Sea *Pocillopora* sp. is dominantly associated with *Symbiodinium* A1, while *Acropora* sp. mainly hosts *Symbiodinium* C41 [63].

## Molecular hydrogen negatively affects the photophysiology of *Acropora* sp. and *P. verrucosa* under ambient seawater temperatures

Contrary to our expectations, the addition of molecular hydrogen at ambient seawater temperatures resulted in a reversed effect in *Acropora* sp., significantly reducing the $ETR_{max}$ and $E_k$. Interestingly, hydrogen at ambient temperatures also led to a significant decrease of the respiration and photosynthesis rates as well as chlorophyll c2 concentrations in *P. verrucosa* together with the lowest Symbiodiniaceae cell densities and chlorophyll a concentrations, although these were not significant.

While supplementation with antioxidants can lead to detrimental effects - even resulting in mortality by potentially disrupting the balance of the antioxidative system [64,65] - hydrogen is generally regarded safe [29,66]. As molecular hydrogen selectively reduces the most harmful ROS and RNS, hydroxyl radicals and peroxynitrite, while not affecting other ROS and RNS that have a potential physiological role, hydrogen is considered as an ideal antioxidant [29].

We therefore propose a potential mechanism by which molecular hydrogen affects *Acropora* sp. and *P. verrucosa* under ambient seawater temperatures, presumably through the inhibition of nitrogen fixation in diazotrophs within the coral holobiont. Several *in vitro* studies have demonstrated that hydrogen competes with atmospheric nitrogen, thereby reducing the formation of bioavailable nitrogen, such as ammonium [67–70]. Under oligotrophic conditions, nitrogen fixation is critical for supplying bioavailable nitrogen [71,72] and therefore the inhibition could limit the overall nitrogen availability of the coral holobiont, particularly under ambient seawater temperatures, ultimately suppressing the photophysiology and growth of the coral holobiont [73–80]. Given the short generation time of Symbiodiniaceae, usually ranging from one to several days [81,82], a decreased growth rate may therefore explain an overall loss of Symbiodiniaceae cells in this short period. This may account for the lowest Symbiodiniaceae cell density and areal chlorophyll a concentration observed in *P. verrucosa* at ambient temperatures when exposed to molecular hydrogen, presumably associated with the significantly reduced photosynthesis rates and areal chlorophyll c2 concentrations. The variations between species might be correlated to differences in nitrogen fixation rates. Since nitrogen fixation rates were ~ 3-fold higher in *Acropora hemprichii* than in *P. verrucosa* [83], a partial inhibition of nitrogen fixation through the addition of hydrogen may have a greater impact on the already lower nitrogen fixation rates of *P. verrucosa*.

This theory is further supported by the temperature-dependent effects of hydrogen maintaining the photosynthesis and respiration rates of *P. verrucosa* under heat stress. At elevated temperatures, nitrogen fixation rates within the coral holobiont were shown to increase [84–86]. Consequently, the potentially reduced nitrogen fixation caused by molecular hydrogen [67–70] might help balance nitrogen availability, resulting in higher productivity compared to hydrogen addition at ambient temperatures. This scenario may even be beneficial under heat stress mitigating the risk of excessive endosymbiont proliferation [55] by maintaining a nitrogen-limited environment essential for the successful coral-algae symbiosis [19,22,86–88].

In contrast to this theory, Rädecker et al. [89] demonstrated that the ammonium produced by the increased nitrogen fixation rates under heat stress was not assimilated by the endosymbionts. Given these circumstances, the inhibitory effect of hydrogen may not support the endosymbionts under heat stress but potentially affect other microbial communities like nitrifiers and denitrifiers [22,90]. However, this theory remains speculative and future studies should address the effects of molecular hydrogen on nitrogen fixation and the bacterial community within the coral holobiont.

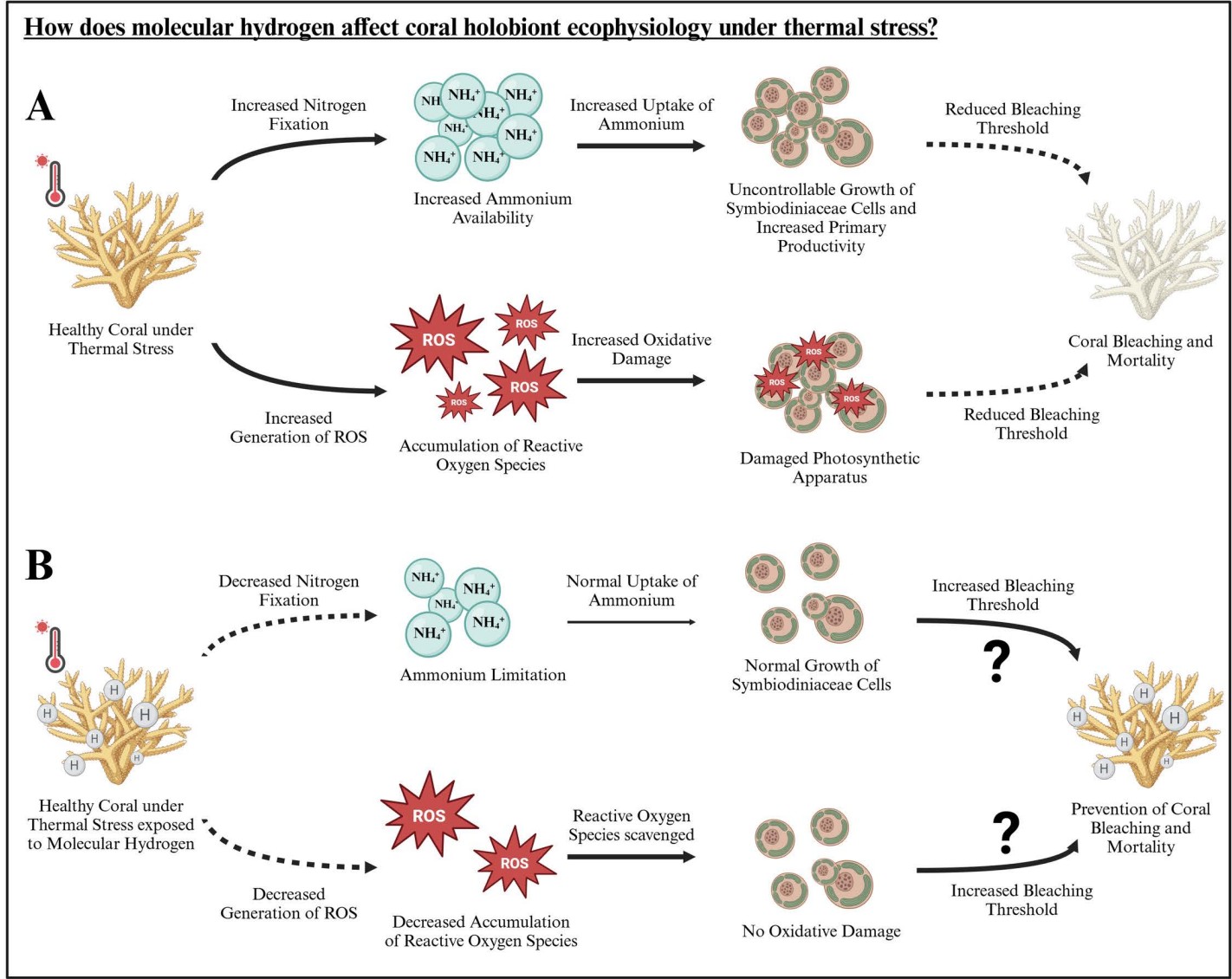

**Fig 6. Hypothesized effect of (A) heat stress and (B) heat stress in combination with molecular hydrogen on the coral holobiont ecophysiology.** First results indicate that hydrogen prevents oxidative damage of the photosystem of the Symbiodiniaceae cells. The potential mechanism of molecular hydrogen behind this effect is illustrated above with dotted lines indicating a decrease and bold lines indicating an increase in the respective process compared to each other. The question mark indicates a potential long-term effect which needs to be addressed in further studies (Illustration created with BioRender.com).

## Ecological implications

Overall, our findings suggest that the effect of hydrogen addition on the coral holobiont is temperature-dependent. While hydrogen exposure had negatively affected the ecophysiology of *Acropora* sp. and *P. verrucosa* at ambient seawater temperatures, it showed a beneficial effect on the symbiont photophysiology of *Acropora* sp. under heat stress, indicating that molecular hydrogen is incorporated and potentially contributes to enhanced coral resilience under heat stress.

It is, however, important to acknowledge the study's limitations. The effects of molecular hydrogen were examined in only two hard coral species in the Central Red Sea and were limited to 48h. Additionally, the experimental setup utilized a single tank for each treatment and species, designed specifically for treatment application rather than extended coral maintenance. While this design allowed precise and consistent environmental conditions and treatment addition, future long-term studies should aim to include an acclimatization phase and a greater number of tanks to mitigate potential tank effects over extended periods. Despite these limitations, our research offers initial evidence of the temperature-dependent effects of molecular hydrogen. Therefore, it can pave the way for future studies especially investigating following research questions: What specific mechanisms underlie molecular hydrogen's role in coral holobionts, particularly in relation to nitrogen fixation and oxidative damage (hypothesized mechanisms in Fig 6)? What are the long-term effects of molecular hydrogen on coral holobionts under both ambient and heat-stressed conditions?

If the hypothesized mechanisms of molecular hydrogen's effects on the coral holobiont are uncovered in future studies, it could offer invaluable insights into the mechanisms underlying coral bleaching potentially supporting the "Oxidative Theory" [23,24]. Further, this could lay the groundwork for the development of new tools in combat against coral bleaching, as ROS and RNS are not only involved in the cascades triggered by heat stress, but also by light stress, cold stress, high nitrate concentrations and bacterial infections [20]. However, careful consideration is needed regarding the safety and potential application of molecular hydrogen.

Given hydrogen's short retention time in water [91], the effects of molecular hydrogen on corals are presumably limited to the period immediately following administration preventing adverse effects at ambient seawater temperatures. Since hydrogen is a very small molecule rapidly diffusing across cell membranes [25], it is very likely that hydrogen will also affect other reef organisms. A recent study, for instance, demonstrated that hydrogen stimulates the growth of diverse marine bacteria [92]. Therefore, the effects of molecular hydrogen on the microbiome as well as other invertebrates and vertebrates, and the underlying mechanisms driving the enhanced thermal resistance, must be clarified prior to application.

With increasing interest in hydrogen as a potential emission-free and renewable energy carrier [26,93], extensive research has focused on developing simple methods for hydrogen production. Electrolysis devices converting distilled water or even seawater into molecular hydrogen are helping to make hydrogen widely available and affordable, thereby enhancing its potential in various applications [94,95]. Given the widespread abundance of hydrogen-producing bacteria [96] and their likely presence within the coral microbiome, identifying these bacteria could further reveal new beneficial microorganisms potentially supporting the development of effective probiotics [97–100].

To explore the full potential of molecular hydrogen, however, further research is essential. In particular, it is important to uncover the underlying temperature-dependent mechanisms, namely those associated with oxidative damage and nitrogen fixation, as well as establishing an optimal hydrogen concentration.

## Supporting information

**S1 File. Supplementary figures and statistical tables.**
(PDF)

**S1 Data. Supplementary data.**
(XLSX)

## Acknowledgments

We would like to thank the whole Marine Microbiomes working group for their great assistance during this work. A big thanks also goes to the boat crew and scientific diving team of the Red Sea Research Center as well as the Coastal and Marine Resources Core Lab of the King Abdullah University of Science and Technology for their support in all marine operations.

## Author contributions

**Conceptualization:** Malte Ostendarp, Mareike de Breuyn, Yusuf C. El-Khaled, Raquel S. Peixoto, Christian Wild.

**Data curation:** Malte Ostendarp.

**Formal analysis:** Malte Ostendarp.

**Investigation:** Malte Ostendarp, Mareike de Breuyn.

**Methodology:** Malte Ostendarp, Mareike de Breuyn, Yusuf C. El-Khaled.

**Resources:** Yusuf C. El-Khaled, Neus Garcias-Bonet, Raquel S. Peixoto, Christian Wild.

**Software:** Malte Ostendarp.

**Supervision:** Yusuf C. El-Khaled, Raquel S. Peixoto, Christian Wild.

**Writing – original draft:** Malte Ostendarp.

**Writing – review & editing:** Malte Ostendarp, Mareike de Breuyn, Yusuf C. El-Khaled, Neus Garcias-Bonet, Susana Carvalho, Raquel S. Peixoto, Christian Wild.

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
