## [Decision Letter · Decision Letter 0]

26 Sep 2024

PONE-D-24-32201Molecular hydrogen can minimize negative effects of heat stress on the hard coral genus *Acropora*PLOS ONE

Dear Dr. Ostendarp,

Thank you for submitting your manuscript to PLOS ONE. After careful consideration, we feel that it has merit but does not fully meet PLOS ONE’s publication criteria as it currently stands. Therefore, we invite you to submit a revised version of the manuscript that addresses the points raised during the review process.

We look forward to receiving your revised manuscript.

Kind regards,

Anderson B. Mayfield, Ph.D.

Academic Editor

PLOS ONE

Journal Requirements:

3. Please expand the acronym “KAUST” (as indicated in your financial disclosure) so that it states the name of your funders in full.

4. Please note that funding information should not appear in the Acknowledgments section or other areas of your manuscript. We will only publish funding information present in the Funding Statement section of the online submission form. Please remove any funding-related text from the manuscript. 

**Additional Editor Comments:**

Hello,

Thank you for submitting your article to PLoS ONE, and I apologize for the length of time it took to have it reviewed. All three reviewers appreciated the novelty of the idea but had issues with the experimental design and interpretation of the data, especially given the overall small scope of the project; the conclusions, then, may not be commensurate with the results (some of which do NOT support the prevailing primary conclusion). One reviewer recommended rejection, but I am hoping that some of the experimental design issues were simply described inadequately. If, however, this reviewer (#1) is correct in their interpretation of how the study was conducted, I may instead recommend that further experimentation be performed, with a later resubmission. As such, I am leaving this open as a "major revision," but if you do not think you can address the reviewers' concerns, I completely understand your decision to either re-submit after further data are collected (or trying your luck with another journal). Hopefully, at a minimum, you find the reviewers' comments useful, and we certainly need new means of fostering coral resilience, either through environmental improvement or boosting of thermotolerance, and so the novel nature of the study is certainly commendable.

Reviewers' comments:

Reviewer's Responses to Questions

**Comments to the Author**

1. Is the manuscript technically sound, and do the data support the conclusions?

Reviewer #1: No

Reviewer #2: Partly

Reviewer #3: Partly

2. Has the statistical analysis been performed appropriately and rigorously? 

Reviewer #1: No

Reviewer #2: Yes

Reviewer #3: Yes

3. Have the authors made all data underlying the findings in their manuscript fully available?

Reviewer #1: Yes

Reviewer #2: Yes

Reviewer #3: Yes

4. Is the manuscript presented in an intelligible fashion and written in standard English?

Reviewer #1: Yes

Reviewer #2: No

Reviewer #3: Yes

5. Review Comments to the Author

Reviewer #1: This manuscript describes intriguing findings with regard to molecular hydrogen mitigating heat stress in Acropora sp. corals. However, I have some fundamental concern regarding the experimental design; specifically lack of replication. As I understand (and perhaps I have misunderstood), each of the four treatments was applied on a separate day, to a separate collection of fragments, in a single aquarium. In this case, although there were six colonies involved, these represent six biological samples, but they are not statistical replicates because they do not represent an independent application of the treatment (the treatments were applied in a single tank). Hence, the application of inferential statistics is not valid (Hurlburt 1984 Ecol Monog 54: 187-211). Meanwhile, I am not a photophysiologist, so cannot comment on the appropriateness of the photophysiological methods/calculations/models.

The target, responsive species in the study is referred to as Acropora spp. Firstly, spp. is a plural notation (i.e. it refers to multiple species). If the authors purport that this sample is a single species (some defense of this point could be appropriate), the correct notation would be ‘sp.’ I understand that Red Sea Acropora may have particular taxonomic challenges, but can any identifying information be given with regard to this responsive sp.? (e.g. growth form, photo,??). Relatedly, since you have tested only one (unnamed) Acropora spp. you should refrain from extrapolating to the whole genus in the title.

Lastly, I found the presentation of the results extremely difficult to follow. Generally, results are presented in a parallel order with the methods and a single statistical test/model is described at the time. I appreciate that the authors are attempting to link the different responses in the results, but it was very hard to follow. Better to order the results talking about one response/test at the time and leave the linkage for the discussion.

Some additional suggestions:

Ln 111: is it possible to do any validation of the manufacturer’s claim in regard to the actual H2 levels? I do not understand the chemistry of how these ‘generators’ work but I am curious how reliable they are . . . .

Ln 138-139: Seems nonsensical to call something dead when it is not dead (i.e. 25% alive). As best I can see, the cited Casey et al. 2015 defined it this way “low mortality was defined as 0–20% loss of tissue from the coral branch, partial mortality was defined as 20–80% loss of tissue from the coral branch, and high mortality was defined as 80–99% loss of tissue from the coral branch” (Casey et al. 2015)

Fig1A: Confused as to the overlapping bars . . I expect the lower portion ® to be the same extent as the non-overlapping portion of the (Pgross) upper bar; but this does not seem to be the case for the first bar (Acropora, Control, 26o). Is this an error?

Also, very confusing to interpret statistical results from this graph, and these statistical results in the text are scattered among different sections . . . .The fig captions says “A significant temperature main effect according to the PERMANOVA with Monte-Carlo test (p < 0.05) is presented with letters (above bar: Pgross, below bar: R) and black lines grouping the temperature conditions” However, I do not see any letters below in panel B, and there are two letters (a and b) depicted above the 26o treatment bars . . . . The text ( Ln 274-276) indicates “Ek was the only parameter to exhibit a significant decrease in both control and hydrogen treatments” implying that R and P (Fig 1B) were not significant. I suggest to include the main effect p-values with the figure rather than rely on lines and letters in such a complicated figure.

-The figures are referenced out of order in the text (part of why the results are confusing)

Reviewer #2: This paper should be commended for its original research on novel mechanisms to support corals coping with thermal stress. In my opinion, it is clear from this work that molecular hydrogen has some effect on some heat stressed corals, and for that reason, is a valuable line of inquiry.

The effects observed with these experiments were somewhat small and the story appears complicated in a way that could be more meaningfully addressed in the discussion. Specifically, addressing the results of each of the measured parameters and why these may have deviated from expectations given the proposed mechanisms of hydrogen enrichment that are offered in the introduction.The discussion also spends too much space discussing potential implementation of molecular hydrogen and not enough discussing the results in detail, in the context of the hypothesized function and mechanisms, and how that might interact with environmental variation in situ. Specific future studies should be proposed to fill in some of the gaps of this work, and build on it.

I would definitely prefer that the experiments ran longer to capture more potential effects of the treatments, with acclimation periods to account for variations in collection timing (or controlling for these analytically, at a minimum), given the incredible sensitivity of these physiological parameters to the environment. That said, these experiments are difficult, labor-intensive, and often logistically-constrained, and I do think the contribution is valuable, if imperfect.

The paper would benefit from technical editing to correct typos, grammatical errors, informal language, acronyms that aren’t introduced, and for clarity and completeness. Choosing standardized language for describing the different controls and treatments would help with clarity in the results and discussion sections.

The specific conclusions are not clear in the discussion, and the conclusion that ‘molecular nitrogen helps corals’ is generalized too broadly there and in the title and abstract. The results only apply to a single species, in the region of the Central Red Sea, and only for one parameter, with a somewhat weak effect. The conclusion should be stated clearly, with clearly delineated assumptions and qualifications about what it might mean, and what limitations the study has for broader interpretation. The explanation related to nitrogen fixation for the difference between hydrogen enrichment on ambient and heat stressed corals makes sense, but is pure speculation and should be identified as such.

I recommend technical editing and a rewrite of the discussion to more completely analyze the results and place them in the context of coral ecophysiology.

Reviewer #3: The authors present a mostly well-designed test of whether molecular hydrogen modulates coral thermotolerance, a novel and interesting question. They find that adding molecular hydrogen to 32°C-treated Acropora increases these corals’ electron transport rates relative to 32°C-treated Acropora without molecular H. They also demonstrate that molecular H decreases the ETR of 26°C-treated Acropora and decreases metabolic rates and chlorophyll concentration for P. verrucosa. The text and figures are clearly presented.

My chief concern about this manuscript is that the overall framing does not adequately capture the data. The title, abstract, and summary model figure (Fig. 6) highlight the heat-mitigatory effects of molecular H but neglect to mention that molecular H seems to decrease photosynthetic health (decreased Ek and ETRmax in Acropora, and lower P and R rates in P. verrucosa) at ambient temperatures. These temperature-dependent responses strike me as fundamental results that are worth highlighting in the abstract and title, as they provide critical context and raise biologically interesting questions – i.e., is there a tradeoff whereby the protective effect of molecular H comes at the cost of dampened photophysiology under ambient conditions? If so, it is imperative that any future study of molecular H consider the potential drawbacks of using this to protect corals. As a result, it feels like claims that molecular H “strengthens” coral physiology (line 412) are too generalized.

Also, actual molecular hydrogen concentrations from the experiment should have been measured and reported with salinity and temperature, since [H2] was the water parameter being manipulated. I understand from the methods as written that the hydrogen water generator manufacturer estimates ~3ppm per cycle for this equipment. However, we don’t know how stable the authors expect that to be during incubation. The concentration could presumably fluctuate based on offgassing, coral or microbial metabolism in the treatment, etc. For the sake of replicability, readers should know the actual [H2]. If these data aren’t currently available, could an additional trial be run with these same [H2] water generators using similar corals under the same conditions to give a sense of the actual incubation [H2]? Can it be estimated somehow? Without being certain whether and to what degree the focal manipulation of this experiment succeeded, it is difficult to put much stock in the conclusions here.

The figures are well-organized and clear, particularly the creative and intuitive display of photosynthetic rate parameters in Figure 1. I always prefer to see individual datapoints layered onto box-and-whisker plots to give readers a better sense of the data distribution, so I would recommend that the authors add these to Figures 1 and 2.

Much of the discussion section about nitrogen limitation is interesting but speculative (lines 374-406), and should be condensed.

If these concerns can be addressed, it is an intriguing result that molecular H can decrease some bleaching signatures under heat stress in Acropora. The conservation applications of these results are likely limited given the difficulty of distributing molecular hydrogen to wild corals covered in the discussion. I find these data more interesting from a mechanistic perspective, as they might provide additional support for an oxidative-stress-centered model of coral bleaching. I think that with a clearer exploration of the temperature-by-treatment interactions observed here, and a more moderated conclusion about potential benefits of molecular H, this paper could be a valuable contribution to the bleaching literature.

6. PLOS authors have the option to publish the peer review history of their article (what does this mean? ). If published, this will include your full peer review and any attached files.

**Do you want your identity to be public for this peer review?** For information about this choice, including consent withdrawal, please see our Privacy Policy .

Reviewer #1: No

Reviewer #2: No

Reviewer #3: **Yes: ** Luella Allen-Waller

---

## [Author Response · Author response to Decision Letter 1]

11 Nov 2024

Dear Reviewers, Dear Dr. Anderson B. Mayfield,

We would like to resubmit our revised manuscript entitled: “Temperature-dependent responses of the hard corals Acropora sp. and Pocillopora verrucosa to molecular hydrogen”. We sincerely appreciate the constructive and valuable feedback from all of you. We have addressed all these comments in the table included in the attached "Response to the Reviewers.docx", highlighting our actions and the corresponding changes made in the manuscript.

Major changes include the revision of the title, abstract, results and discussion section to more accurately reflect the temperature-dependent effects of molecular hydrogen adjusting the overall conclusion. Additionally, we have provided a statement regarding the crucial concerns of the experimental design justifying our analysis, while also clearly stating these limitations within the manuscript. Overall, we believe that both the clarity and coherence of the entire manuscript have been enhanced, resulting in a more comprehensive presentation of our findings.

Yours sincerely on behalf of all co-authors,

Malte Ostendarp

---

## [Decision Letter · Decision Letter 1]

17 Dec 2024

PONE-D-24-32201R1Temperature-dependent responses of the hard corals *Acropora* sp. and *Pocillopora verrucosa* to molecular hydrogenPLOS ONE

Dear Dr. Ostendarp, Thank you for submitting your manuscript to PLOS ONE. After careful consideration, we feel that it has merit but does not fully meet PLOS ONE’s publication criteria as it currently stands. Therefore, we invite you to submit a revised version of the manuscript that addresses the points raised during the review process.

We look forward to receiving your revised manuscript.

Kind regards,

Anderson B. Mayfield, Ph.D.

Academic Editor

PLOS ONE

Journal Requirements:

**Additional Editor Comments:**

Thank you for your patience. I have now had this article reviewed by another two reviewers, one of whom endorsed it in full and the second with only some minor comments. As such, I believe this article is nearing the finish line. I would, however, urge you to ensure that all concerns of the very first reviewer from the first round of review were thoroughly addressed, as they were quite severe (pointing to some potentially critical experimental design flaws). If that reviewer's interpretation of the design was correct, then the article may ultimately have a diminished impact. Just a thought since, as someone employing coral reef interventions myself, we want to ensure that these methods are robust before they start spreading life wildfire!

Reviewers' comments:

Reviewer's Responses to Questions

**Comments to the Author**

1. If the authors have adequately addressed your comments raised in a previous round of review and you feel that this manuscript is now acceptable for publication, you may indicate that here to bypass the “Comments to the Author” section, enter your conflict of interest statement in the “Confidential to Editor” section, and submit your "Accept" recommendation.

Reviewer #4: (No Response)

Reviewer #5: (No Response)

2. Is the manuscript technically sound, and do the data support the conclusions?

Reviewer #4: Partly

Reviewer #5: Yes

3. Has the statistical analysis been performed appropriately and rigorously? 

Reviewer #4: Yes

Reviewer #5: Yes

4. Have the authors made all data underlying the findings in their manuscript fully available?

Reviewer #4: Yes

Reviewer #5: Yes

5. Is the manuscript presented in an intelligible fashion and written in standard English?

Reviewer #4: Yes

Reviewer #5: Yes

6. Review Comments to the Author

Reviewer #4: I was invited to review the second, revised version of this manuscript. The authors did a good job addressing previous reviewers’ comments. However, I still have some comments and suggestions on how the manuscript could be improved.

Methodology

The authors presume that readers are familiar with the terms and biological relevance of various physiological traits connected to photosynthesis and they do not explain or discuss them in the paper. Since I am not an expert on photosynthesis, I found it challenging to follow the results and discussion when terms like P gross, P net, respiration, ETRmax and Ek are discussed within the context of nutrient shuffling and coral-algal symbiosis maintenance. It would enhance understanding if the authors explained what these traits represent as proxies and how their shifts are relevant to the biology of coral thermal resilience.

Figures

Figure 3 and Figure 4 labels don’t correspond to their respective mentions in the text. I didn’t find any mention/explication of figure 3A, while figure 4A is mislabeled as ETRmax and 4B as Ek in the text (line 360 in the tracked-changes version).

Discussion

I am not sure I follow the authors’ deduction that, based on their results, they believe molecular hydrogen prevents coral bleaching through its competition with nitrogen. My understanding is that the coral host limits nitrogen bioavailability to the symbiont to reduce its population growth. Although the doubling time of symbionts in adult coral tissue is debated, the general consensus is that it occurs over days rather than hours, as far as I know (and I might be wrong). So it is already challenging to discuss the effect on population growth based on the experiment that lasted only 48 hours.

Moreover, the data presented show that hydrogen addition had no effect on the coral respiration (Fig 1), photosynthetic efficiency (Fig 2), Symbiodiniacae density, or chlorophyll content, but it did affect ETRmax (Fig 3) in the Acropora species. In Pocillopora corals, hydrogen addition affected coral respiration (Fig 1) and chlorophyll content (Figure 5) with no effect on photosynthesis efficiency (Fig 2) or symbiont cell density. It’s two different effects and as much as I understand differences between coral species, I don’t find that there is enough evidence in the discussion right now to claim that these changes are caused by a difference in nitrogen bioavailability to the symbiont. Could authors discuss detailed evidence from other literature that connects these traits to nitrogen availability? How do they reconcile the differences between species while drawing a general model from the data? (Figure 6)

On the same note, the authors claim that hydrogen addition presumably prevents/mitigates coral bleaching. However, I do not see evidence in the presented data supporting this statement. Where in the data do they demonstrate that molecular hydrogen mitigates coral bleaching?

Furthermore, they discuss the potential strategies on how to use molecular hydrogen on reefs to mitigate damage during acute thermal stress. For example, the sentence “Primarily, molecular hydrogen should only be administered during heat stress events with hydrogen gas concentrations lower than 4% to minimize the risk of explosion…” (line 545) reads like a direct instruction for field application. This is concerning, as there is no definitive proof in this study that molecular hydrogen has an effect on coral thermal resilience or mortality. Additionally, the long-term effects and trade-offs of molecular hydrogen on coral health and reef ecosystems remain unknown but can be detrimental. Suggesting untested interventions could be dangerous, especially if individuals with good intentions but limited expertise act on this advice.

I believe scientific community should be very careful about how we communicate our results to a broad audience, particularly when proposing strategies to mitigate reef damage under climate change. Only scientifically validated methods should be proposed with such direct wording in scientific literature.

Despite these concerns, I commend the authors for highlighting the potential of molecular hydrogen in studying coral-algal symbiosis maintenance and disruption. Understanding the cellular and molecular principles of coral bleaching and resilience is crucial for developing effective reef protection and restoration strategies. Every promising approach should be carefully and comprehensively studied before being proposed for real-world application. The authors offer new and potentially powerful perspectives on coral reef protection, and their work should be published after addressing these points.

Reviewer #5: This manuscript aims to address a key outstanding question in coral ecophysiology of the cellular mechanisms of coral bleaching and offers insights into theoretical interventions for bleaching-mitigation. I commend the simultaneous use of multiple methods including algal symbiont flow cytometry, quantification of symbiont chlorophyll content, measuring photochemical efficiency and quantifying photosynthesis:respiration ratios to assess holobiont heat tolerance. Together, results provide multiple complementary lines of evidence to assess the interacting effects of short-term heat stress and molecular hydrogen on photosynthetic & photochemical efficiency in the coral holobiont.

I agree with previous reviewers that re-framing the manuscript around the temperature-dependence of the effect of hydrogen on the heat stress response better supports the results reported and has improved the manuscript. The manuscript is generally logically organised, with detailed and thorough descriptions of most methods and clear figures which are intuitive to interpret and aesthetically consistent throughout the manuscript. I would recommend the addition of the following to the introduction and discussion sections to prepare this manuscript for publication: Firstly, the relevance of the metrics reported in the results to the ‘temperature tolerance’ of the holobionts needs to be prefaced in the introduction before the reader can relate results to the hypothesis. Secondly, additional summarising of results is needed in the discussion to more comprehensively interpret the nuances of how heat and hydrogen affected various facets of photochemical performance in each species tested.

Specific comments (ordered by line number)

• 57: More detail on the oxidative theory of coral bleaching would be helpful here. As would brief mention of other prominent mechanistic theories for coral bleaching. eg the shift from nitrogen-limitation to carbon-limitation (especially since you address nitrogen limitation in your discussion).

• 76: The final paragraph in the introduction would benefit from more setup to prepare the reader for the results they’re about to see (specifically, the relationships between photosynthesis, respiration, photochemical efficiency, ETRmax, Ek and chlorophyll, and how these metrics relate to ‘temperature tolerance’).

• 97: Since fragments for molecular hydrogen treatments were collected from your colonies after fragments for control treatments, it’s possible that the wound-healing responses triggered by earlier collections affected coral responses to hydrogen treatments- worth addressing this somewhere.

• 133: In light of the unintended temperature variation, it would be helpful to see the per-treatment temperature profiles visualised in your supplemental materials.

• 228: How did you account for between-colony differences across these results? i.e. did you include colony effects in models?

• 268: No explanation in the methods for why several colony replicates were excluded from two of the treatments.

• 353: At the start of the discussion, it would be helpful to have a summary of the main results and how the different metrics complement/oppose each-other with respect to the effects of heat and hydrogen on temperature tolerance in these two species.

• 379: Lacking some comment on possible reasons why Acropora & Pocillopora responses differed.

• 416: Comment on symbiont taxa commonly associated with your study species in reefs near to your collection site, since this would surely affect the metrics recorded in this study relating to photosynthesis.

7. PLOS authors have the option to publish the peer review history of their article (what does this mean? ). If published, this will include your full peer review and any attached files.

**Do you want your identity to be public for this peer review?** For information about this choice, including consent withdrawal, please see our Privacy Policy .

Reviewer #4: **Yes: ** Eva Majerova

Reviewer #5: No

---

## [Author Response · Author response to Decision Letter 2]

27 Jan 2025

Dear Dr. Anderson B. Mayfield, Dear Reviewers,

We would like to resubmit our revised manuscript entitled: “Temperature-dependent responses of the hard corals Acropora sp. and Pocillopora verrucosa to molecular hydrogen”. We sincerely appreciate the constructive and valuable feedback from you. We have addressed all the comments, as outlined in the response to the reviewers document, detailing the actions taken and the corresponding changes made to the manuscript.

Minor revisions include updates to the introduction and discussion sections together with the reference list to provide a clearer explanation of the response parameters, particularly regarding coral thermal tolerance and their interdependence. Overall, we believe that the manuscript's clarity, especially concerning the response parameters, has improved, while minimizing potential misunderstandings of the ecological implications.

Yours sincerely on behalf of all co-authors,

Malte Ostendarp

---

## [Editor Report · Decision Letter 2]

11 Feb 2025

Temperature-dependent responses of the hard corals *Acropora* sp. and *Pocillopora verrucosa* to molecular hydrogen

PONE-D-24-32201R2

Dear Dr. Ostendarp,

We’re pleased to inform you that your manuscript has been judged scientifically suitable for publication and will be formally accepted for publication once it meets all outstanding technical requirements.

Kind regards,

Anderson B. Mayfield, Ph.D.

Academic Editor

PLOS ONE

Additional Editor Comments:

Hello,

Thank you for your patience. Despite some polarizing reviews at the outset (which then led me to try and find totally different ones for the second round of reviews), I believe this article is now suitable for publication in PLoS ONE. Thank you for your patience.

---

## [Editor Report · Acceptance letter]

PONE-D-24-32201R2

PLOS ONE

Dear Dr. Ostendarp,

I'm pleased to inform you that your manuscript has been deemed suitable for publication in PLOS ONE. Congratulations! Your manuscript is now being handed over to our production team.

Kind regards,

on behalf of

Dr. Anderson B. Mayfield

Academic Editor

PLOS ONE